Towards the development of cost-effective point-of-care diagnostic tools for poverty-related infectious diseases in sub-Saharan Africa

Ofori Benedict 1 2
Twum Seth 2
Nkansah Yeboah Silas 1 2
Ansah Felix 1 2
Amofa Nketia Sarpong Kwabena 1 2 kansarpong@ug.edu.gh
1 West African Centre for Cell Biology of Infectious Pathogens, University of Ghana, Legon , Accra , Ghana
2 Department of Biochemistry, Cell and Molecular Biology, College of Basic and Applied Sciences, University of Ghana, Legon , Accra , Ghana
Gillespie Joseph
Electronic publication date: 2024 Jun 21
Publication date: 2024
Volume: 12
Electronic Location ID: e17198
Received 2023 Aug 21; Accepted 2024 Mar 14
Copyright: © 2024 Ofori et al.
Copyright year: 2024
Copyright holder: Ofori et al.
License: This is an open access article distributed under the terms of the Creative Commons Attribution License, which permits unrestricted use, distribution, reproduction and adaptation in any medium and for any purpose provided that it is properly attributed. For attribution, the original author(s), title, publication source (PeerJ) and either DOI or URL of the article must be cited.
License URL: https://creativecommons.org/licenses/by/4.0/

Keywords: Resource-limited settings, Diagnostic tools, Point-of-care testing (POCT), Diagnosis, Sub-Saharan Africa (sSA), Poverty-related infectious diseases (PRIDs)

Funding: DELTAS Africa DEL-15-007: Awandare Welcome Trust DEL-15-007 UK Foreign, Commonwealth & Development Office This work was supported by a DELTAS Africa grant (DEL-15-007: Awandare). This research was funded in whole by the Welcome Trust (DEL-15-007) and the UK Foreign, Commonwealth & Development Office, with support from the Developing Excellence in Leadership, Training and Science in Africa (DELTAS Africa) programme. The funders had no role in study design, data collection and analysis, decision to publish, or preparation of the manuscript.

==============================
In this review, we examine the current landscape of point-of-care testing (POCT) diagnostic tools designed for poverty-related infectious diseases (PRIDs) in sub-Saharan Africa (sSA) while delineating key avenues for future advancements. Our analysis encompasses both established and emerging diagnostic methods for PRIDs, addressing the persistent challenges in POCT tool development and deployment, such as cost, accessibility, and reliability. We emphasize recent advancements in POCT diagnostic tools as well as platforms poised to enhance diagnostic testing in sSA. Recognizing the urgency for affordable and widely accessible POCT diagnostic tools to detect PRIDs in sSA, we advocate for a multidisciplinary approach. This approach integrates current and emerging diagnostic methods, explicitly addressing challenges hindering point-of-care (POC) tool development. Furthermore, it recognizes the profound impact of misdiagnosis on public and global health, emphasizing the need for effective tools. To facilitate the successful development and implementation of POCT diagnostic tools in sSA, we propose strategies including the creation of multi-analyte detection POCT tools, the implementation of education and training programs, community engagement initiatives, fostering public-private collaborations, and the establishment of reliable supply chains. Through these concerted efforts, we aim to accelerate the development of POCT in the sSA region, ensuring its effectiveness and accessibility in addressing the diagnostic challenges associated with PRIDs.

Introduction

Accurate laboratory data and interpretation play a key role in achieving a robust healthcare system with precise diagnosis and treatments (Mrazek et al., 2020; Shaw, 2016). Point-of-care testing (POCT), which refers to conducting laboratory procedures near the patient rather than in a central facility, is essential for accurate diagnosis and monitoring in clinical settings, particularly in sub-Saharan Africa (sSA), where infectious diseases are the leading cause of deaths (Dima, 2021; Salyer et al., 2017).

Infectious diseases such as HIV, malaria, and tuberculosis (TB) are responsible for a significant number of deaths in sSA. However, the lack of adequate laboratory testing modalities raises uncertainty regarding the accuracy of these estimates (Petti et al., 2006). The latest available data indicate that HIV-related deaths in sSA in 2022 were 380,000, TB-related deaths in 2016 were 417,000, and malaria-related deaths in 2022 were 580,000 (Africa CDC, 2023; UNAIDS, 2023; World Health Organization, 2023).

POCT is well-suited for sSA in the diagnosis of poverty-related infectious diseases (PRIDs) compared to non-point-of-care (POC) tests. Its significance is underscored by its compatibility with readily collected sample types, particularly evident during the COVID-19 pandemic when sSA often faced challenges with laboratory infrastructure and skilled personnel. (Baldeh et al., 2023).

Unlike non-POC tests, POCT eliminates the need for extensive laboratory infrastructure and skilled personnel, provides affordable and reliable tests, and can be implemented within sustainable financing models, making it a pragmatic choice for improving diagnostic capabilities in sSA (Moetlhoa et al., 2023; Pai et al., 2012).

PRIDs predominantly affect impoverished communities with limited access to healthcare, clean water, sanitation, and proper nutrition (Makoge et al., 2017; WHO, 2012). Neglected tropical diseases (NTDs), a subset of PRIDs, disproportionately affect those living in poverty, especially those residing in proximity to infectious vectors and domestic animals (Engels & Zhou, 2020). Consequently, early and accurate diagnosis becomes imperative, as witnessed in recent epidemics and pandemics like Ebola, Lassa fever, yellow fever, Zika, and COVID-19, where the absence of local diagnostic capacity has had significant consequences.

The 2022 Ebola outbreak reported 170 cases and 61 deaths (Centre for Diseases Control, 2023). A case in Ivory Coast in 2021 initially tested positive for Ebola, leading to a large contact-tracing operation and a vaccination campaign. However, a new lab analysis from France later revealed that the patient did not have Ebola, highlighting the potential impact of misdiagnosis on public health efforts (Al Jazeera, 2022).

Lassa fever continues to impose substantial clinical, economic, and health security burdens in Nigeria, which bears the highest burden of the disease (Wada et al., 2022). In Nigeria alone, Lassa fever has infected 3,897 people and caused 1,319 deaths, including 71 healthcare workers, with a case fatality rate of 33.8% for confirmed cases (Wada et al., 2022). In the West African region, the long-standing aftereffects of Lassa fever, such as sensorineural hearing losses, are causing a significant social and economic burden, as stigmatization and isolation lead to an increase in unemployment and depression rates (Aloke et al., 2023). These figures underscore the broader impact of inadequate diagnostic capacity, emphasizing the urgent need for improved testing capabilities at the local level.

It is essential to focus on the development of affordable diagnostic tools tailored for deployment in remote and rural areas to improve patient outcomes and control the spread of these diseases. Funding from organizations like the Point-of-Care Technologies Research Network and the European and Developing Countries Clinical Trials Partnership is instrumental in supporting research on diagnostic tools and technologies to improve the performance of diagnosing PRIDs and should be actively sought. Despite the support from such organizations, many POCT tools remain prohibitively expensive for widespread usage in sSA, and their reliability is often uncertain (Frank et al., 2019; Nijhuis et al., 2018). Challenges, such as the lack of skilled personnel, inadequate supply chains, and poor health infrastructure, further impede the development and the use of POCT diagnostic tools in the region (Kuupiel et al., 2019; Shaw, 2016). Existing diagnostic tools for PRIDs in sSA are often centralized laboratory-based assays that require specialized equipment and trained personnel. This renders them expensive and accessible only in urban areas with modern health facilities. The complexity and time-consuming nature of these tools pose significant challenges for implementation in resource-limited settings.

While several studies have focused on the cost-effectiveness of diagnostic tools (De Broucker et al., 2021; Moetlhoa et al., 2023; Sharma et al., 2021), none have specifically focused on strategies to expedite the development of POCT tools in sSA. This review aims to fill this gap by introducing innovative approaches that could catalyze the creation of cost-effective POCT tools in the region. Given the escalating costs of POCTs in sSA, our objective is to propose novel and inventive solutions that effectively address the current challenges in the development process.

We strongly believe that the limited availability of accurate and affordable diagnostic tools, in addition to the high prevalence of PRIDs, underscores the urgent need for cost-effective and accessible POCT diagnostic tools. This review targets a diverse audience, including healthcare professionals who utilize POCs for patient care, researchers focusing on PRIDs, and governmental and non-governmental policymakers shaping healthcare strategies in sSA. The insights presented in this review intend to inform strategic decision-making, foster collaboration, and inspire targeted efforts toward the development and implementation of effective POCT diagnostic tools for PRIDs in sSA.

Methodology

Literature search and data sources

The literature search was conducted through PubMed (https://pubmed.ncbi.nlm.nih.gov/), Google Scholar (https://scholar.google.com/), and ScienceDirect (https://www.sciencedirect.com/). Boolean operators (AND, OR) and predetermined keywords related to Poverty-Related Infectious Diseases (PRIDs) and Point-of-Care Testing (POCT) in case studies for disease diagnosis were employed. The search terms included variations such as (Infectious diseases) AND (sub-Saharan Africa) OR (Infectious Disease) AND (Poverty), (Diagnosis) AND (Resource-Limited Settings), (Infectious Disease) AND (Point-of-care testing tools), (Hepatitis) OR (Malaria) OR (Tuberculosis) OR (HIV) AND (Negative predictive value) OR (Positive predictive value) AND (sub-Saharan Africa). We restricted the search to publications released between January 1995 and December 2023. Supplementary data sources were retrieved through Google, contributing to a comprehensive and inclusive exploration of relevant literature beyond the primary databases initially mentioned.

Specific eligibility criteria guided the selection of article titles for our review. The exclusion process involved a two-step approach. Initially, records were excluded if they did not specifically focus on PRIDs in sSA. Following this, from the obtained results, a second round of exclusion was conducted to filter out studies that did not emphasize the implementation of diagnostic tools, with a particular focus on point-of-care diagnostic tools. The intention was to gather insights into the diagnostic accuracy, advantages, and challenges associated with POCT tools, recognizing that understanding unsuccessful implementations is equally informative.

The flow chart (Fig. 1) illustrates the systematic process employed for the selection of pertinent studies, focusing on PRIDs and POCT in resource-limited settings.

Figure 1 Flow chart of manuscripts reporting on PRIDs and their implementation for diagnosis in sSA, with a particular emphasis on point-of-care diagnostic tools.

Electronic literature search between 1995 and 2023 was performed using specific keywords, and inclusion and exclusion criteria to identify manuscripts that met our inclusion criteria.

Overview of poverty-related infectious diseases in sub-saharan africa

Various socioeconomic factors exacerbate PRIDs, contributing to significant morbidity and mortality rates in the sSA region. Addressing the challenges posed by PRIDs in sSA necessitates the implementation of comprehensive and sustainable strategies to enhance healthcare access and public health interventions.

Risk factors, prevalence, and incidence of poverty-related infectious diseases in sub-Saharan Africa

In 2020, the global tally of malaria cases reached approximately 200 million, resulting in nearly 500,000 annual deaths. Of these fatalities, over 90% occurred in sSA, with a pronounced impact on young children (Stonely, 2023; Centre for Diseases Control, 2021). According to the Global HIV Statistics Facts Sheet 2023 by UNAIDS, in 2022, approximately 4,000 new HIV infections were reported weekly among adolescent girls and young women aged 15-24 globally, with 3,100 of these cases occurring in sSA (UNAIDS, 2023).

In Africa, the true extent of PRIDs like TB and NTDs such as human African trypanosomiasis, schistosomiasis, lymphatic filariasis, and Guinea worm disease cases remains uncertain due to inadequate laboratory and diagnostic facilities. Several factors impact the prevalence, incidence, and risk factors of PRIDs in sSA. Cultural beliefs, low socioeconomic levels, and restricted access to healthcare all contribute to the region’s high rate of PRID prevalence. Managing these risk factors and being aware of their effects is crucial if sSA is to effectively prevent and manage PRIDs. POCT plays a pivotal role in bridging these gaps by offering immediate and accurate diagnostic results at the point of care and overcoming these challenges. It is crucial to recognize that PRIDs in sSA not only pose public health challenges but also hinder the region’s growth and progress toward achieving Sustainable Development Goals (Shears, 2007). Resolving poverty and infectious diseases are intertwined goals and addressing one can positively impact the other.

Inverse relationship between economic challenges and accessibility and affordability of point-of-care testing in sub-Saharan Africa

Africa boasts a rich landscape teeming with diverse and valuable mineral resources, contributing nearly one billion metric tons of minerals valued at $406 billion in 2019. However, despite this wealth, these resources are not harnessed to enhance diagnostic capacity in the region (Al Jazeera, 2022). The impact of poverty on the burden of PRIDs in sSA varies geographically (Boutayeb, 2010).

Regarding diagnostic costs for malaria, studies have revealed that the average cost of rapid diagnostic tests (RDT) diagnosis stands at USD 2.19 per test, encompassing material costs (USD 1.51) and labor costs (USD 0.68) (Du et al., 2020). In comparison, the average cost of microscopy diagnosis is USD 6.98 per test, inclusive of material costs (USD 0.18) and labor costs (USD 6.80). World Bank data indicates that most people in endemic areas of sSA live on less than 2.15 dollars a day, which is even below the average cost of diagnosing prevalent infectious diseases in the region.

While the poverty headcount rate in sSA slightly decreased from 38.9 to 38.3 between 2018 and 2019, the population of individuals experiencing poverty has risen, reaching 424 million in 2019 from 420 million in 2018 (Castaneda Aguilar et al., 2022). To tackle this challenge, POCT emerges as a strategic approach, providing a cost-effective and immediate diagnostic solution aligned with the economic constraints and healthcare disparities prevalent in sSA (Parkes-Ratanshi et al., 2019). By reducing the costs associated with traditional diagnostic methods and offering rapid and accurate results at the point of care, POCT becomes a vital tool in the fight against PRIDs in sSA. The top four countries (Nigeria, Uganda, the DR Congo, and Mozambique) that account for the most malaria deaths in sSA are shown in Fig. 2.

Figure 2 The top four countries account for the most malaria deaths worldwide (Nigeria, DR Congo, Uganda, and Mozambique).

Map created at https://www.mapchart.net/ under a Creative Commons Attribution-ShareAlike 4.0 International License (CC BY-SA 4.0).

The impact of governance structures and resource allocation on diagnostic capacity in sub-Saharan Africa

It is imperative to spotlight the role that poor governance structures play in facilitating the spread of PRIDs for a comprehensive understanding of regional differences and the influence of poverty on PRIDs in sSA. In contrast to most high-income countries, the governments of many African nations opted for relatively earlier implementation of lockdowns on public life during the COVID-19 pandemic, based on the number of confirmed cases. For instance, all African nations except Burundi closed their schools by March 31, 2020, even when several had no verified cases (Günther et al., 2022). The incapacity of medical infrastructure to handle severe COVID-19 cases and the inadequacy of health systems were pivotal factors contributing to the early and often stringent government regulations in sSA.

Effectively managing and treating PRIDs, particularly NTDs, demands substantial financial resources (Boutayeb, 2010). Tuberculosis has the highest prevalence in Southern and Eastern Africa, while malaria is most prevalent in West and Central Africa (Ayles, Mureithi & Simwinga, 2022; Korzeniewski, Bylicka-Szczepanowska & Lass, 2021). Regional variations among sSA countries stem from multiple factors, including disparities in socioeconomic levels, environmental conditions, and healthcare access. Insufficient resources for disease prevention and control compound the impact of poverty on the burden of PRIDs in the region.

Enhancing sSA’s diagnostic capacity can be facilitated through effective tax collection, a significant source of government revenue. Despite geographical variations, certain universal strategies can mitigate the impact of PRIDs, including improving access to healthcare services and advancing health education. The World Health Organization’s 2011 report, ‘State of Health Financing in the African Region,’ assessed how well sSA governments allocate resources to improve healthcare access, revealing varying levels of satisfaction across the region: West Africa at 31.5%, East and Southern Africa at 45.1%, and Central Africa at 30.9%.

Analyzing annual total revenue offers insights into the economic impact of different countries. Figure 3 depicts the total revenue figures for selected countries in 2020 based on revenue generation, with South Africa leading, followed by Nigeria and Kenya. The disjunction between high total revenue and the disproportionate prevalence of malaria-related deaths, as seen in Fig. 2, underscores challenges in resource allocation within the healthcare system in sSA.

Figure 3 Total revenue generated in 2020 for a diverse set of African countries, including both high and low-income countries.

Data obtained from OECD/AUC/ATAF (2022), Revenue Statistics in Africa 2022, OECD Publishing, Paris.

Low-to-middle-income countries must prioritize the inclusion of diagnostics in national health insurance schemes funded by tax revenues to reduce out-of-pocket payments for patients and improve access to diagnostic services (Bigio et al., 2023). The observed discrepancy between robust financial resources and insufficient diagnostic capabilities during disease outbreaks in sSA likely results from systemic inefficiencies and a potential misalignment in healthcare initiative prioritization. This misalignment is reflected in the higher incidence of malaria-related deaths, emphasizing the urgent need for a targeted and strategic redirection of resources to fortify diagnostic capacities.

Current and emerging diagnostic methods for poverty-related infectious diseases

While various diagnostic methods are available for PRIDs in sSA, each method comes with its own set of advantages and disadvantages. The selection of a specific diagnostic method depends on factors such as the type of disease, the availability of laboratory facilities and trained personnel, and cost considerations (Srivastava et al., 2018).

According to MARKETSANDMARKETS (2022), a Pune-based company in India, the global point-of-care diagnostics market was projected to reach a value of $45.4 billion in 2022, with a compound annual growth rate (CAGR) of 0.7% from 2022 to 2027. The substantial growth in this market is fueled by the increasing prevalence of PRIDs, notably evident in sSA. Notably, the development of these POCT tools is primarily led by companies located outside of sSA.

Extensive literature details the advantages and limitations of current diagnostic tools used for PRID diagnosis. In the context of sSA settings, we present the following as the current methods for diagnosing PRIDs, along with their respective advantages and limitations:

Nucleic acid-based diagnostics

Polymerase chain reaction (PCR) is a nucleic acid-based diagnostic method commonly utilized for detecting PRIDs. PCR is known for its exceptional sensitivity and specificity in identifying specific DNA or RNA sequences of pathogens in bodily fluids or blood. However, its use requires specialized laboratory equipment and trained personnel, resulting in higher costs compared to microscopy and RDTs. The PCR technique involves amplifying or directly detecting targeted pathogen genetic material (Gupta, 2019). Its benefits include the ability to detect low levels of pathogens during early-stage infections due to its high sensitivity and specificity.

Microscopy-based diagnostics

Microscopy-based diagnostics have been a longstanding method for diagnosing PRIDs such as malaria and tuberculosis (Lee et al., 2023; Tangpukdee et al., 2009). These techniques involve the visual examination of samples and have been widely accessible and relatively inexpensive compared to PCR-based detection systems. However, accurate interpretation of microscopy data requires skilled personnel due to its lower sensitivity and specificity in contrast to PCR-based methods, particularly in low-density infections (Lee et al., 2023; Morgan et al., 1998).

The challenge lies in the fact that not all healthcare facilities in sSA are equipped with microscopes and trained personnel, creating hurdles in accurately identifying pathogens, especially given the diverse species causing infections like malaria (Sitali et al., 2019). Accurate speciation is needed for early diagnosis and appropriate treatment, as different species may require distinct therapeutic approaches (Berzosa et al., 2018). It is worth noting that misreporting of Plasmodium knowlesi infections as other species have been observed globally when relying solely on light microscopy (Yegneswaran, Alcid & Mohan, 2009).

The advantages of microscopy-based diagnostics include affordability, direct visualization, and morphological detection, while limitations encompass reduced sensitivity, limited specificity, reliance on sample quality, and difficulty in detecting non-viable pathogens.

Rapid diagnostic tests

RDTs are extensively employed for diagnosing PRIDs such as malaria, HIV/AIDS, and NTDs like Buruli ulcer. These tests utilize lateral flow immunoassays to detect specific antigens or antibodies in bodily fluids (Koczula & Gallotta, 2016). RDTs offer several advantages, including simplicity, point-of-care functionality, and rapid results, often available within minutes. They are user-friendly, require minimal training, and can be used in resource-limited settings like sSA where well-equipped laboratory facilities may be scarce.

However, it is important to acknowledge that RDTs have limitations, including lower sensitivity and specificity when compared to nucleic acid-based tests. False positive and negative results are more prevalent, especially in cases of low-density infections.

Imaging-based diagnostics

Medical imaging techniques, such as X-rays and ultrasounds, are frequently employed in the diagnosis of PRIDs such as tuberculosis and NTDs. These imaging-based diagnostics rely on non-invasive techniques to visualize anatomical structures and detect pathological changes in tissues. The advantages of this approach lie in its non-invasiveness and the ability to provide detailed information about the extent and severity of the disease. Additionally, imaging techniques can be used to monitor disease progression and assess treatment responses.

However, it is important to note that these methods have limitations. They can be relatively expensive and require specialized equipment and trained personnel for accurate interpretation. Furthermore, their sensitivity and specificity may be limited, particularly in early-stage infections (Kumar et al., 2008).

Emerging methods for diagnosis of PRIDs

Promising advancements in the field of diagnostics for PRIDs involve the use of microfluidic devices and biosensors. Biosensors employ portable instruments to detect specific biomarkers in bodily fluids, providing a simple and efficient diagnostic approach (Senf, Yeo & Kim, 2020). On the other hand, microfluidic devices offer automation and simplification of complex laboratory procedures, making them suitable for implementation in resource-limited settings like sSA (Gonzalez-Suarez et al., 2022).

An emerging trend in diagnostic technologies is the use of aptamers as biorecognition elements. Aptamers, which are nucleic acids or peptides capable of binding to target molecules such as proteins or small molecules, show potential as effective and affordable POC diagnostic tools (Sarpong & Datta, 2012). They offer advantages over conventional diagnostic techniques, including affordability, stability, and reproducibility. Given the increasing prevalence of PRIDs in sSA, the development of aptamer-based diagnostic assays has the potential to greatly improve diagnostic capabilities in the region and enhance POCT.

The loop-mediated isothermal amplification (LAMP) assay is an increasingly popular molecular diagnostic technique for the detection of PRIDs. It offers several advantages over other methods, such as PCR. LAMP is characterized by its simplicity, rapidity, sensitivity, specificity, and cost-effectiveness (Soroka, Wasowicz & Rymaszewska, 2021). In LAMP, a specific target DNA sequence is amplified using a DNA polymerase and four to six primers under isothermal conditions (Obura et al., 2011; Seevaratnam et al., 2022). Unlike PCR, LAMP does not require a thermocycler and can be performed using a simple water bath or heating block. LAMP has demonstrated successful application in the diagnosis of various infectious diseases, including tuberculosis, malaria, dengue fever, Zika virus, and COVID-19 (Amaral et al., 2021; Garg, Ahmad & Kar, 2022). Figure 4 presents an overview of the various types of POCT tools currently available. As shown in Fig. 4, these technologies include lateral flow assays, microfluidic devices, nucleic acid amplification tests, and various biosensors.

Figure 4 Point-of-care diagnostics for PRIDs.

Point-of-care diagnostics and importance

Healthcare professionals can use POCT devices to diagnose a wide range of medical conditions including biomarkers of infectious diseases at or near the point of care (Chen et al., 2019; Nichols et al., 2007). This positively impacts the management of patients presenting with infectious disease symptoms. In 2010, the World Health Organization endorsed the first POCT device, GeneXpert, for detecting HIV-associated tuberculosis. This device integrated sample preparation, nucleic acid amplification, and detection into a single mechanism, completing nucleic acid detection within 2 h. Subsequently, several other POCT devices have been developed for diagnosing PRIDs.

The LIAT analyzer (IQuum) was designed for HIV detection (Tanriverdi, Chen & Chen, 2010), Truelab Uno® for malaria detection (Nair et al., 2016), and GenePoC for Influenza A and B virus detection (Stevens et al., 2017), amongst others. With the recent global outbreak of infectious diseases, especially the COVID-19 pandemic, there has been a surge in research focusing on the development of POCT devices. Consequently, a new series of POCT devices have been specifically designed for detecting SARS-CoV-2 (Chan et al., 2020).

Role of POCT devices in the early detection and treatment of PRIDs

POCT devices enable real-time results, allowing public health officials to swiftly identify the specific pathogen causing an outbreak. This enables the implementation of targeted measures to control its spread, thereby limiting the potential for larger epidemics or pandemics (Hanafiah, Garcia & Anderson, 2013). These devices provide crucial insights for guiding appropriate therapy, especially in infectious diseases that require specific treatments like antimicrobial therapy. POCT tools are invaluable for tracking treatment responses, particularly in persistent PRIDs such as HIV and tuberculosis (Jani & Peter, 2013; Tucker, Bien & Peeling, 2013). They facilitate the accurate diagnosis of diseases and aid in determining the optimal treatment strategy.

Enhancing accuracy in point-of-care diagnostics for poverty-related infectious diseases in sub-saharan africa

We analyzed selected studies focusing on malaria, hepatitis, tuberculosis, and HIV to assess the positive predictive value (PPV) and negative predictive value (NPV) of POC tools. Table 1 presents an overview of the PPV and NPV obtained from selected studies from our search results, highlighting the use of POC tools for diagnosing PRIDs in sSA. PPV represents the proportion of true positives among individuals who test positive for the disease, while NPV represents the proportion of true negatives among individuals who test negative. Factors such as disease prevalence, POC test kit quality, healthcare worker expertise, and result interpretation influenced the diagnostic accuracy of the POC test kits identified in our search. In sSA, PPV and NPV serve as critical criteria to evaluate the effectiveness of POC diagnostic tools for disease diagnosis. The attention given to PPV and NPV of POC diagnostic tools in sSA is essential for several reasons, as discussed in the next section.

Table 1 PPV and NPV of some selected studies on PRID diagnosis in sSA.

Article reference source	Disease condition	Test kit	Type of test	PPV	NPV	
Sitali et al. (2019)	Malaria	SD Bioline malaria Ag Pf, Standard Diagnostics Inc., Republic of Korea	Rapid diagnostic test	75.9%	94.1%	
Parsel et al. (2017)	Malaria	FAST Malaria stain kit (427760, QBC Diagnostics Inc., Port Matilda, PA) and ParaLens AdvanceTM microscope attachment (QBC Diagnostics Inc., Port Matilda, PA) compared with traditional Giemsa Stain	Microscopy	27.99% for FAST Malaria stain kit and 19.62% for traditional Giemsa Stain	95.28% for FAST Malaria stain kit and 95.01% for traditional Giemsa Stain	
Sahle et al. (2017)	Tuberculosis	Determine TB LAM Ag strip (Alere international ltd, South Africa)	Rapid diagnostic test	86.7%	79.4%	
Galiwango et al. (2014)	HIV	PIMA CD4 analyzer	Rapid diagnostic test	87.5%	94.9%	
Ndoye et al. (2022)	HIV	Xpert® HIV-1 Qual Assay (Cepheid) and m-PIMA™ HIV-1/2 Detect Assay (Abbott)	Rapid diagnostic test	100% for both kits in agreement to reference platform (Cobas AmpliPrep/Cobas TaqMan (CAP/CTM)	100% in agreement to reference platform (Cobas AmpliPrep/Cobas TaqMan (CAP/CTM)	
Mossoro-Kpinde et al. (2022)	Multiplex tool for Hepatitis C and B, and HIV 1 and 2	Exacto® Triplex HIV/HCV/HBsAg (Biosynex, Strasbourg, France)	Rapid diagnostic test	100%	100%	

It is imperative to prioritize POCT tools demonstrating high PPV and NPV for accurate diagnosis and management of PRIDs. Suboptimal PPV and NPV of these diagnostic tools pose a risk of false positives and negatives, potentially leading to misdiagnosis.

To enhance PPV and NPV, we propose the following strategies:

Combining two or more diagnostic tests that target various parts of the target pathogen to improve the accuracy of diagnosis.

Enhance sample collection and handling quality

Educating healthcare personnel, such as phlebotomists, in the collection and handling of samples correctly can help reduce the incidence of false positives and negatives.

Evaluating the POC tool’s performance in the local population

The performance of the POC tool may differ depending on the population being assessed. Assessing the tool’s performance in the local community can assist in uncovering any factors that may affect the accuracy of the diagnosis and improve the test’s PPV and NPV.

Employ quality control measures

Quality control procedures, such as regular calibration and maintenance of POC tools can increase the test’s accuracy and reliability resulting in improved positive and negative predictive values.

Challenges affecting the development and implementation of poct in sub-sahara africa

POCT tools play a vital role in rapidly and effectively diagnosing various PRIDs. However, the development and implementation of these POCT tools face numerous challenges in sSA. It is crucial not to overlook the necessity of developing POCTs in this setting, considering the significant public health concerns associated with the disease burden of PRIDs, such as NTDs. sSA countries encounter several challenges in developing and implementing new POCT tools to enhance diagnosis. These challenges include poverty, limited infrastructure, high disease burden, disparities in the healthcare system, and regulatory and ethical considerations, among others. This section outlines the major challenges hindering the development of POCT tools in sSA.

Poverty

Approximately 51% of the population in sSA lives on less than $1.25 per day, with around 73% surviving on less than $2.00 per day, underscoring pervasive poverty levels (Hotez & Kamath, 2009; Nwani & Osuji, 2020; Zhang et al., 2022). Poverty in the region manifests through inadequate sanitation, social exclusion, limited access to quality healthcare, lack of clean water, and malnutrition (Mara et al., 2010). These conditions contribute to various morbidities including PRIDs discussed earlier in this review. An estimated 475 million people, constituting 36% of the total population in sSA, live in poverty and are affected by these diseases (Aikins & McLachlan, 2022), underscoring the urgent need for the development of diagnostic tools for early detection and treatment. Despite the progress made in combating these diseases through the development of vaccines and new drugs, the development of POCT tools in the region faces persistent barriers due to low productivity. Financial support and collaborations from foreign aid, non-governmental organizations, corporate bodies, and the government are essential for overcoming these challenges. Substantial resources, particularly financial ones, are requisite for POCT tool development, and without adequate support, the process becomes difficult to sustain. Moreover, for these tools to be effective, they must be affordable and accessible to individuals, especially those in rural areas. However, given the prevailing poverty levels, achieving this goal would be nearly impossible without adequate support and aid.

Limited infrastructure

The absence of well-established local POC manufacturing biotech companies in sSA poses a hindrance to development across various sectors, despite numerous aid and financial interventions by governments and corporate institutions. This deficiency in POC tool manufacturing in sSA significantly impacts various processes, including the development and implementation of new POC diagnostics. The region faces a significant shortage of research-intensive laboratories compared to other parts of the world. The design and development of POCT tools require substantial resources and the right equipment, which are often lacking due to limited infrastructure. Essentials like electricity, internet connectivity, and databases for data storage are not readily available, making it challenging to operate the equipment necessary for POC tool development (Lakmeeharan et al., 2020).

The issue of electricity poses a major challenge to the region’s development. World Bank projections indicate that approximately 50.6% of the sub-Saharan population has access to electricity, with rural areas being particularly underserved. Insufficient investment in the power sector, poor infrastructure, limited power generation capacity, population growth, and governance issues contribute to this problem (Hafner, Tagliapietra & de Strasser, 2018). Even in regions with easy access to electricity, frequent blackouts and power outages are common. This poses a significant obstacle to POC tool development, as most equipment and materials rely on electricity. For example, the LAMP assay, an emerging diagnostic tool, requires the enzyme Bacillus stearothermophilus DNA Polymerase I, which needs to be stored at −20 °C for stability and function (Obura et al., 2011; Seevaratnam et al., 2022). Working with such reagents becomes costly and resource-draining due to the unreliable electricity supply in the region.

Sustainability and scalability

To ensure the successful development and implementation of POCT devices, sustainability and scalability must be prioritized. Sustainability, in this context, refers to the long-term functionality of POCT tools (Palamountain et al., 2012). Considering the challenges faced by healthcare facilities in sSA, including infrastructure limitations and personnel shortages, sustainability becomes a critical factor. Moreover, the cost of POCT tools is a significant consideration in a region affected by poverty. Affordability is essential to ensuring the accessibility of POCT tools across healthcare facilities, regardless of their financial resources. POCT tools should be designed to be durable, easy to maintain, and require minimal specialized training to operate effectively (Palamountain et al., 2012).

With a population estimated to be around 1.18 billion in 2021 and a growth rate of 2.6%, population size is a key factor for POCT tool developers to consider. Meeting the needs of such a large and growing population requires scalability. POCT tools should be designed to be easily transportable and capable of functioning in various climatic conditions to ensure scalability (Palamountain et al., 2012). Establishing effective supply chain measures is also crucial for achieving scalability.

Regulatory and ethical considerations

Regulatory and ethical considerations play an important role in the successful development and deployment of POCT devices. Approval from regulatory bodies, including Standards Authorities and Health Ministries, is essential to ensure the safety, effectiveness, and applicability of these tools before they can be utilized by the population. However, the regulatory processes can be protracted, causing unwarranted delays. Furthermore, these processes vary across countries and regions within sSA, making it challenging for suppliers and developers to bring their products to market. Even if a test passes all regulatory requirements and is deployed, it may still fail clinically in terms of sensitivity and specificity. This can occur because the clinical performance of a test is often based on data generated from different geographical locations and may not accurately represent the population in sSA, leading to significant differences in test response (Palamountain et al., 2012; Yager, Domingo & Gerdes, 2008).

To gather research data on the sensitivity and specificity of these devices, study participants must be recruited. The recruitment processes aim to obtain informed consent while ensuring participant safety and privacy, reflecting ethical considerations. However, sSA is a region with diverse cultural beliefs and languages, which can present barriers to effective communication between recruiters and participants as well as obtaining informed consent. This poses a significant challenge to the development of POCT tools in the region.

Health system challenges

The sub-Saharan African region lacks robust healthcare plans and systems to effectively address the burden of PRIDs. The inadequacy of infrastructure and the absence of proper health systems emerge as significant obstacles to the development and deployment of diagnostic tools. Limited funding for healthcare research in the region further constrains resources available for the development of POCT tools, presenting hurdles in the creation and evaluation of new technologies. Consequently, there is a critical need for increased investments in healthcare infrastructure and research to confront the prevalent disease burden in sSA and foster advancements in the development of these tools.

Potential solutions for developing and implementing poc diagnostic tools in sub-saharan africa

The successful implementation of POCT tools over the years has been impeded by various challenges, including a lack of local capacity, inadequate supply chains, and limited community engagement. Drawing from existing literature, we offer suggestions and ideas to address these obstacles and enhance POC tool development in sSA. Figure 5 highlights key areas that should be considered to improve patient health outcomes in the region.

Figure 5 Effective methods for the development and implementation of POC tools in sSA.

Education and training programs

The effective utilization of POCT tools relies on the expertise of clinicians, technicians, and patients, making education and training programs essential for their successful implementation. These programs are designed to equip healthcare professionals with the requisite skills to operate, maintain, and interpret results from POCT diagnostic instruments, integrating them seamlessly into standard care practices Creating awareness of the importance of POCT diagnostics through educational initiatives can foster their widespread adoption (Carter et al., 2011). Laboratory technicians need to have a comprehensive understanding of the biochemical principles underlying laboratory tests to ensure accurate interpretation of results and ensure appropriate diagnosis. Regular training and continuing education programs should be provided by laboratories to strengthen technicians’ foundation in laboratory testing principles. This knowledge is essential to prevent misdiagnosis and potential harm to patients (M.V et al., 2018; Doust & Glasziou, 2013). Laboratory technicians, often college graduates with backgrounds in medical laboratory science (MLS) in sSA, require a comprehensive understanding of biochemical principles for accurate result interpretation and diagnosis. Regular training and continuing education programs are crucial to fortify technicians’ foundations in laboratory testing principles, ensuring the prevention of misdiagnosis and potential harm to patients. In this context, ongoing education and training become imperative for clinical laboratory professionals, enabling them to stay abreast of the latest advances in laboratory science, technology, and regulations. Continuous learning ensures their competency, acquisition of new skills, and adaptation to changes in the field, allowing them to deliver high-quality services while adhering to regulatory and accrediting standards.

Public-private partnerships

Public-private partnerships (PPPs) have emerged as a promising yet underexplored avenue in the development of cost-effective POCT tools. However, PPPs have gained recognition among national and international stakeholders as a practical approach to enhancing the efficiency of health systems and improving health outcomes (Ravishankar & Lehmann, 2015). By leveraging the resources, technical expertise, and infrastructure of both sectors, PPPs can support the development and utilization of POCT tools. Private-sector businesses can contribute by developing and manufacturing POCT instruments, while public-sector organizations can facilitate their distribution and utilization. Collaborative efforts between governments, organizations, and businesses can ensure the availability of POCT tools to the populations that need them the most. Moreover, the potential of PPPs to accelerate the creation of affordable POCT tools deserves further exploration. By combining the resources and expertise of the public and private sectors, PPPs have the potential to reduce the development costs of POCT tools while ensuring their accessibility and sustainability. The National Policy on Public-Private Partnership, as outlined in the 2011 report by the Ministry of Finance, Ghana, emphasizes the importance of promoting PPP as a strategy to combine public and private resources (Ministry of Finance and Economic Planning, 2011). Additionally, PPPs can facilitate technology transfer, which is crucial for enhancing regional capacity in POCT tool production and maintenance (UNCTAD, 2014). Nonetheless, the success of PPPs relies on careful planning, open communication, and trust among partners. It is important to identify and address potential bottlenecks such as conflicts of interest, inadequate regulation, and power imbalances to ensure the effectiveness of PPPs.

Development of appropriate supply chains

The establishment of efficient and reliable supply chains is paramount for the successful implementation of POCT tools, ensuring their timely and cost-effective delivery to remote and underserved areas. Addressing logistical challenges associated with supply chain management has led to innovative strategies, including drone delivery, mobile clinics, and telemedicine (Jeyabalan et al., 2020). To maintain the proper functioning of POCT devices, it is vital to establish regular and timely supply chains for consumables and replacement components (Dima, 2021). Notably, initiatives like the USAID Global Health Supply Chain Program in have demonstrated improved delivery of patient diagnostic samples and results, including viral load, early infant diagnosis, and tuberculosis, using drones (USAID, 2020). The Ghana Drone Delivery Service, launched in 2019, further showcases the potential of drones in improving supply chain efficiency and reaching remote and underserved areas (WHO Africa, 2019). While drone delivery services in sSA have primarily focused on emergency services like blood pints and drugs, the potential use of drones for delivering POCT tools to resource-limited settings is gaining attention. Establishing drone delivery systems may pose challenges, as observed in negotiations between the Ghanaian government and a US company specializing in on-demand drone delivery (BBC, 2018). Nevertheless, integrating innovative approaches like drone delivery alongside traditional methods such as road transport holds promise in promptly improving access to POCT tests for diverse populations in the region.

Community engagement and involvement

Active community engagement is pivotal for the successful implementation of POC diagnostic techniques. Through engagement with communities, trust can be built, and awareness of these tools can be increased. In settings where there is a shortage of professional healthcare workers, community members can be trained to operate POCT equipment. This requires collaboration between the ministries of health, local governments, and healthcare facilities in the sub-Saharan African region to educate and provide informal training to community members on the usage of POCT kits. Local governments can incentivize healthcare officers to lead sensitization programs, fostering community engagement and reducing congestion in healthcare facilities for basic tests that could be easily conducted at home with accessible and user-friendly POCT kits. By involving communities in the utilization of POCT tools, the overall healthcare system can be strengthened and decentralized, allowing for efficient and timely testing, particularly in underserved areas.

Developing cost-effective POCT tools with multiplexing capabilities

Many currently available POCT kits are singleplex, designed to test for one pathogen at a time. This approach can be expensive and time-consuming, particularly when dealing with patients presenting multiple conditions. In cases like HIV/AIDS, multiple screening procedures are typically necessary, involving initial screening tests for HIV antibodies followed by confirmatory testing such as Western blot or ELISA assays (Urio et al., 2015). To address this issue, multiplexing techniques have been developed, allowing for the simultaneous testing of multiple pathogens in a single sample. Multiplexing can significantly reduce diagnostic costs and time. Various methods, such as microarrays, flow cytometry, and PCR-based assays, can be used to achieve multiplexing. Previous studies have demonstrated the successful development of multiplex POCT for infectious disease detection, cancer diagnosis, and drug monitoring. An example of multiplex POCT technology is the BioFire FilmArray System, which utilizes multiplex PCR-based technology to detect up to 22 respiratory infections, including SARS-CoV-2, in a single test. By incorporating multiplexing capabilities into low-cost POCT tools, access to healthcare can be increased in limited settings by reducing both diagnostic costs and time. The continuous advancements in multiplexing technologies are expected to drive the development of more advanced and efficient POCT kits in the future.

Integrate innovative biotechnology to advance POCT in sSA

The integration of innovative biotechnologies, such as the hand-powered paper centrifuge, holds great promise for advancing POCT in sSA. A potential benefit of innovative biotechnology in advancing POCT in sSA is easy accessibility. Hand-powered devices do not rely on electricity, making them suitable for areas with unreliable electricity. This ensures that even remote health facilities without access to a constant electricity supply can perform essential diagnostic tests without dependence on external resources. Aside from accessibility, compact and portable biotechnological tools are easy to transport, enabling healthcare to reach rural locations where traditional laboratory equipment may be impractical. This facilitates on-the-spot testing, reducing the need for patients to travel from rural to urban centers for diagnostic needs.

The hand-powered paper centrifuge and similar technologies enable quick processing of biological samples, resulting in faster test results for timely interventions, particularly in cases where immediate treatment decisions can significantly impact patient outcomes (Li et al., 2020). The introduction of innovative biotechnologies not only brings advancements in diagnostics but also plays a pivotal role in local capacity building. This builds a skilled workforce capable of utilizing advanced technologies for improved patient care and diagnostics.

Addressing cultural barriers

Cultural beliefs and traditions can have a big impact on healthcare-seeking behavior, acceptance, and use of medical services and diagnostic technologies. As a result, it is critical to understand local ideas and values and to create culturally suitable technologies. Here are some ways to address cultural barriers:

Local communities should be involved in the development and deployment of POCT tools. Interact with community leaders, health professionals, and traditional healers to learn about their healthcare beliefs, practices, and expectations. This will aid in the development of diagnostic tools that are both culturally suitable and accepted.

Utilize local languages to convey the aim and benefits of point-of-care diagnostic equipment. This will help to create trust and raise the likelihood of the diagnostic instruments being accepted and used.

Integrate local traditions, such as traditional medicine, into the design and implementation of POCT tools. This will help to bridge the gap between modern healthcare and traditional methods and boost diagnostic tool acceptance.

Address stigma

Address stigma associated with specific diseases, such as HIV/AIDS, which may hinder people from accessing treatment and using diagnostic equipment. Create ways to combat stigma and discrimination, as well as to increase acceptance and use of diagnostic instruments.

Educate healthcare professionals

To increase cultural awareness and competency, educate healthcare providers about local cultural beliefs and practices. This will help to ensure that healthcare providers are respectful of and responsive to their patient’s cultural needs.

Conclusion

POCT devices offer significant advantages in the detection of PRIDs, particularly in resource-limited settings. They provide portability, speed, and accuracy, enabling healthcare practitioners to make informed decisions quickly and improve patient outcomes while reducing the spread of infectious diseases. However, several challenges hinder the widespread use of POCT devices in such settings. Addressing these challenges requires a multifaceted approach. Education and training programs are crucial to equip healthcare professionals with the necessary skills to operate and interpret results from POCT devices. Public-private collaborations can facilitate the development of low-cost POC instruments, combining the resources and expertise of both sectors. Developing multiplex POCT tools capable of simultaneously detecting multiple pathogens can reduce diagnostic costs and time, enhancing healthcare accessibility in low-resource settings. Furthermore, establishing reliable supply chains is imperative to ensure the timely and dependable delivery of diagnostic tools to remote and underserved areas. By focusing on education, collaboration, multiplexing technology, and supply chain development, we can overcome the barriers that hinder the utilization of POCT devices in low-resource settings. These efforts will contribute to improved healthcare delivery, better patient outcomes, and a reduced burden of infectious diseases in resource-limited settings.

Additional Information and Declarations

Competing Interests

Author Contributions

Data Availability

The authors declare that they have no competing interests.

Benedict Ofori conceived and designed the experiments, performed the experiments, analyzed the data, prepared figures and/or tables, authored or reviewed drafts of the article, and approved the final draft.

Seth Twum performed the experiments, analyzed the data, prepared figures and/or tables, authored or reviewed drafts of the article, and approved the final draft.

Silas Nkansah Yeboah performed the experiments, analyzed the data, authored or reviewed drafts of the article, and approved the final draft.

Felix Ansah analyzed the data, authored or reviewed drafts of the article, and approved the final draft.

Kwabena Amofa Nketia Sarpong conceived and designed the experiments, analyzed the data, authored or reviewed drafts of the article, and approved the final draft.

The following information was supplied regarding data availability:

This is a literature review.

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
