# Peer review of "Towards the development of cost-effective point-of-care diagnostic tools for poverty-related infectious diseases in sub-Saharan Africa"

_PeerJ, doi:10.7717/peerj.17198_

## Round 0.1 · original submission · Major Revisions

· Academic Editor

Major Revisions

Dear Dr. Ofori and colleagues:

Thanks for submitting your manuscript to PeerJ. I have now received two independent reviews of your work, and as you will see, the reviewers raised some concerns about the manuscript and scope of the review. Despite this, these reviewers are optimistic about your work and the potential impact it will have on research studying infectious disease diagnosis in sub-Saharan Africa. Thus, I encourage you to revise your manuscript, accordingly, considering all the concerns raised by the two reviewers.

Please edit the manuscript for clarity and typos.

Please ensure that your figures and tables contain all the information that is necessary to support your findings and observations. Also, several methodological approaches need more explanation.

Good luck with your revision,

-joe

**Language Note:** The review process has identified that the English language must be improved. PeerJ can provide language editing services - please contact us at [email protected] for pricing (be sure to provide your manuscript number and title). Alternatively, you should make your own arrangements to improve the language quality and provide details in your response letter. – PeerJ Staff

Reviewer 1 ·

Basic reporting

-Line 42: "--" may be a typo.
-Lines 64 to 74: The paragraph discussing malaria and the average cost of RDT should be relocated to later sections, possibly in section 3.2. The current placement disrupts the flow of the review and creates a disconnect between the preceding and succeeding paragraphs.
-Line 75: "While several studies have focused…," it is essential for the authors to provide references to support their statement.
-Lines 123-125: "According to ……cases occurring in sSA," please include a reference link.
-Line 173: "According to……," please provide an accessible link or source to support the information.
-Lines 252-258: The paragraph belongs more appropriately in the method section.
-I recommend swapping Section 5 and Section 6 to improve the overall reading flow.
-Lines 408, 415, and 419: The observed glitches might be attributed to the conversion of the Word file to PDF. Ensure that the document formatting is correct.
-Lines 463-465: “Regarding the statement about Ghana Drone Delivery… and underserved areas”. Please provide accessible resources or references to support the information.

Experimental design

-Lines 99-101: “In addition to employing renowned databases like PubMed, Google Scholar, and ScienceDirect, we expanded our coverage by utilizing other relevant search engines”. It is not clear what “other relevant search engines” are.
-Lines 103-106: “By combining a wide array of search terms and utilizing multiple reputable search engines, we aspired to provide a comprehensive exploration of the topic, focusing on the development of cost-effective diagnostic tools for infectious diseases of poverty.” I suggest relocating these sentences into later sections, rather than in the method section.
- I recommend that the authors also discuss the potential benefits of innovative biotechnology in advancing POCT in sSA, such as the hand-powered paper centrifuge.

Validity of the findings

-Figure 1: Inconsistency between lines 73-74 and the legend and figure caption should be addressed (“the number of malaria cases” in lines 73-74 vs. “malaria deaths” in the legend and figure caption). Also, I recommend including the exact number on the top of the figure, particularly for these four countries.
- Section 3.2: I agree with the authors' points; however, the combination of Figure 1 and Figure 2 somewhat challenges the central argument presented in Section 3.2. Figure 2 shows that Nigerians have a relatively high total revenue, but Nigeria is also among the top four countries with the highest malaria-related deaths, as indicated in Figure 1. The authors should provide a more comprehensive discussion on this matter.
-Figure 2: I recommend adjusting the Y-axis scale (e.g. max is 100000) and moving "(Million USD)" to the Y-axis. Furthermore, provides clear justification for the inclusion of specific countries in the figure. If these countries were chosen due to the availability of data, consider rephrasing "Selected" to explicitly communicate this rationale for clarity.

Additional comments

NA

Reviewer 2 ·

Basic reporting

Firstly, I commend the authors on their ambitious broad and cross-disciplinary coverage of the topic of infectious disease diagnosis in sub-Saharan Africa, including challenges and opportunities for improvement. The review is applicable to a wide audience, including health professionals, health managers, and policy makers.

The introduction could be strengthened by providing more specific information in key places. For example, even if the estimates are uncertain due to inadequate laboratory testing methodology as you state, what are the best estimates of HIV, TB, etc. death burden (lines 40-42)? What magnitude of consequences did the Ebola, etc. epidemics have (lines 48-49)? What is the burden is misdiagnosis? I would suggest that you choose a few key figures to demonstrate your point in these cases.

It is not immediately clear why POCT is best suited for infectious diseases. You explain further in to the introduction that accurate diagnostic tools need to be available in rural settings and state that POCT tools are essential to "improving testing, medical diagnosis, monitoring, and reducing misdiagnosis in the region" (line 43). While I fully agree that POCT are suited to infectious disease diagnosis in rural settings, non-POC tests can also be affordable and accurate. I suggest you restructure carefully to improve the argument flow throughout the introduction.

PRIDs not defined in main text (line 44). Spelling this out in the abstract is not sufficient.

What is the denominator for "poverty headcount" (line 70)?

Experimental design

The methodology is not adequately explained. Not all search term combinations include testing/diagnostics, confusing the reader in terms of the aim of the review.

Line 100: "we expanded our coverage by utilizing other relevant search engines." Which ones?

The total number of articles identified seems very low. A Pubmed search of "Infectious disease" AND "sub-Saharan Africa" yields over 2000 articles in the timeframe specified. My suggestion would be to provide a flow chart showing how many articles were identified in the initial search, how many articles were included after applying the selection criteria, how many articles were excluded with reasons for exclusion, etc. to increase transparency.

Why was successful implementation required as inclusion criteria (line 111)? Wouldn't unsuccessful implementation if the reasons for failure were adequately address be informative to the aim of this review?

The second mention of a search (line 252) is confusing. The intention to cover this topic in sufficient detail to conduct a separate search is not indicated in the preceding text and there are no results of the search mentioned later on in the article.

Validity of the findings

Unfortunately, the ambitiousness of the authors to address so many topics in one paper means it falls short on the several of the individual arguments. Many of the arguments are not fully developed and there is often repetitive information, though many good points are made throughout. Furthermore, there are several paragraphs that tackle multiple topics - the clarity of the paper could significantly be improved by focusing each paragraph on a single topic/argument. I would suggest the authors carefully restructure the manuscript to improve the argument flow by making one argument at a time and more fully developing each of them. It is difficult to provide more specific comments before this, however, here are a couple illustrative examples:

1) the paragraph starting at line 121 discusses lack of adequate healthcare, poor sanitation and access to clean water, health education, and nutrition. While all of these things share a common feature, i.e. poverty, the connection with POCT in the context of infectious disease diagnosis is not made.

2) the paragraph starting at line 144 starts out providing background information on Africa's mineral resources (what is the relevance here?), then moves on to government structures, financial resources, regional variations in disease burden, then suggests effective tax collection, access to health care and health education (also mentioned in the paragraph above) as improvement strategies, and finally provides data on health satisfaction with no further comment. This is a lot of information for the reader without a clear story of the point the authors are trying to make.

Line 151-152: developed nations with good infrastructure do not necessarily implement effective control strategies.

---

## Round 0.2 · Minor Revisions

· Academic Editor

Minor Revisions

Dear Dr. Ofori and colleagues:

Thanks for revising your manuscript. The reviewers are very satisfied with your revision (as am I). Great! However, there are a few issues to entertain. Please address these ASAP so we may move towards acceptance of your work.

NOTE: your manuscript has many grammatical and typographical errors. Please enlist the help of some English experts to improve the quality of the writing and presentation.

Best,

-joe

**Language Note:** The Academic Editor has identified that the English language must be improved. PeerJ can provide language editing services - please contact us at [email protected] for pricing (be sure to provide your manuscript number and title). Alternatively, you should make your own arrangements to improve the language quality and provide details in your response letter. – PeerJ Staff

Reviewer 1 ·

Basic reporting

The authors have answered all my questions.

Experimental design

no comment

Validity of the findings

no comment

Reviewer 2 ·

Basic reporting

line 50: It sounds like you are saying “POCT eliminates the need for…affordable and reliable tests, and sustainable financing models”. I would suggest something like, “POCT eliminates the need for extensive laboratory infrastructure and skilled personnel, provides affordable and reliable tests, and can be implemented within sustainable financing models, …”

line 235: Why introduce the abbreviation “HICs” when you don’t use it anywhere else in the paper?

line 104: “…, none of these studies have focused on strategies that can be adopted to accelerate the development of POCT tools in the sSA African region.” What about other studies?

Experimental design

Thank you for including the flow diagram – this is helpful. However, I find it difficult to believe that >27,000 records were screened if the search was conducted in December 2023. Perhaps the search strategy needs to be narrowed to retrieve more relevant articles. The single reason for exclusion in the flow diagram is related to article relevance. Were there no other reasons for excluding articles?

Validity of the findings

Line 45: “POCT is well-suited…” should start a new paragraph

Line 54: “PRIDs predominately affect…” should start a new paragraph

Section “3.1 Risk factors, prevalence, and incidence of poverty-related infections diseases in sub-Saharan Africa” contains information that does not fit under this heading and is repetitive information also contained in the introduction (line 189-201).

Section “3.2” again heading does not match the content of the text.

Please carefully consider which text belongs in each section. There are several instances throughout where the content of the section does not match the heading or even matches a different heading.

It is still unclear to me what is results of the systematic review and what is background information.

Additional comments

It is difficult to find the relevant section without updated line numbers. Please refer to line numbers from the revised version in your responses in the future.

There are still several typos and grammatical mistakes throughout. Please check carefully.

---

## Round 0.3 · accepted · Accept

· Academic Editor

Accept

Dear Dr. Ofori and colleagues:

Thanks for revising your manuscript based on the concerns raised by the reviewers. I now believe that your manuscript is suitable for publication. Congratulations! I look forward to seeing this work in print, and I anticipate it being an important resource for groups studying infectious disease diagnosis in sub-Saharan Africa. Thanks again for choosing PeerJ to publish such important work.

Best,

-joe